# A Compositional Model to Predict the Aggregated Isotope Distribution for Average DNA and RNA Oligonucleotides

**DOI:** 10.3390/metabo11060400

**Published:** 2021-06-18

**Authors:** Annelies Agten, Piotr Prostko, Melvin Geubbelmans, Youzhong Liu, Thomas De Vijlder, Dirk Valkenborg

**Affiliations:** 1Data Science Institute, UHasselt—Hasselt University, Agoralaan 1, BE 3590 Diepenbeek, Belgium; annelies.agten@uhasselt.be (A.A.); piotr.prostko@uhasselt.be (P.P.); melvin.geubbelmans@student.uhasselt.be (M.G.); 2Interuniversity Institute for Biostatistics and Statistical Bioinformatics (I-BioStat), Agoralaan 1, BE 3590 Diepenbeek, Belgium; 3Chemical & Pharmaceutical Development & Supply, Janssen Research & Development, Turnhoutseweg 30, BE 2340 Beerse, Belgium; YLiu186@its.jnj.com (Y.L.); TDEVIJLD@its.jnj.com (T.D.V.)

**Keywords:** DNA, RNA, oligonucleotide, prediction, isotope distribution, mass spectrometry, software

## Abstract

Structural modifications of DNA and RNA molecules play a pivotal role in epigenetic and posttranscriptional regulation. To characterise these modifications, more and more MS and MS/MS- based tools for the analysis of nucleic acids are being developed. To identify an oligonucleotide in a mass spectrum, it is useful to compare the obtained isotope pattern of the molecule of interest to the one that is theoretically expected based on its elemental composition. However, this is not straightforward when the identity of the molecule under investigation is unknown. Here, we present a modelling approach for the prediction of the aggregated isotope distribution of an average DNA or RNA molecule when a particular (monoisotopic) mass is available. For this purpose, a theoretical database of all possible DNA/RNA oligonucleotides up to a mass of 25 kDa is created, and the aggregated isotope distribution for the entire database of oligonucleotides is generated using the BRAIN algorithm. Since this isotope information is compositional in nature, the modelling method is based on the additive log-ratio analysis of Aitchison. As a result, a univariate weighted polynomial regression model of order 10 is fitted to predict the first 20 isotope peaks for DNA and RNA molecules. The performance of the prediction model is assessed by using a mean squared error approach and a modified Pearson’s χ^2^ goodness-of-fit measure on experimental data. Our analysis has indicated that the variability in spectral accuracy contributed more to the errors than the approximation of the theoretical isotope distribution by our proposed average DNA/RNA model. The prediction model is implemented as an online tool. An R function can be downloaded to incorporate the method in custom analysis workflows to process mass spectral data.

## 1. Introduction

Nucleic acids play a pivotal role in the regulation of numerous cellular processes and act as carriers in the storage and processing of genetic information. Therefore, they probably are the most intensively studied biopolymers. Ever since the introduction of ‘soft’ ionisation techniques like electrospray ionisation (ESI) and matrix-assisted laser desorption ionisation (MALDI), the mass spectrometry (MS)-based analysis of nucleic acids and oligonucleotides has received a considerable amount of attention. Several reviews have described the progress in this field over the years [1,2,3]. Despite this progress, the field has matured slowly compared to that of MS-based proteomics. One particular reason for this might be the lack of dedicated and suitable bioinformatics solutions, which has hampered analysis of large amounts of complex data [4,5,6].

The qualitative and quantitative analysis of structurally modified DNA [7] and RNA [8] has become an important application in MS-based analysis of nucleic acids due to its importance in epigenetic and posttranscriptional regulation and for the detection of chemical DNA damage in the form of DNA adducts [9]. As there is an increased interest in how these DNA/RNA modifications impact and regulate the epitranscriptome, more MS and MS/MS-based tools are being developed to characterise these modifications [10,11,12].

Further, synthetic oligonucleotides are an emerging class of therapeutic modalities that might be capable of targeting many proteins (through their expression) that are considered ‘undruggable’ via classical small molecule therapeutics. These oligonucleotide-based therapeutics encompass a diverse group including single stranded antisense oligonucleotides (ASOs), micro-RNAs, aptamers, CRISPR (clustered Regularly Interspaced Short Palindromic Repeats) guide RNAs and double stranded small interfering RNAs (siRNAs). These compounds are generally produced via a solid-phase chemical synthesis approach and vary in size from ~20–30-mers ASOs and siRNAs to ~100-mers for CRISPR guide RNAs. A recent and comprehensive overview on the applications of MS in the field of therapeutic oligonucleotides was published by Pourshaian [13]. It is important to note that, due to their high molecular weight and chemical complexity, intact mass analysis by high-resolution (HR) MS is often a pivotal technique in the identity testing package of therapeutic oligonucleotides for clinical/commercial release of drug substance and drug product [14].

Another class of nucleic acid-based molecules that recently gained momentum in the pharmaceutical industry, mostly as a consequence of the COVID-19 pandemic and subsequent rapid vaccine development, are mRNAs. As mRNA-based therapeutics/vaccines are much larger in size (~1000-mers and more), they are more difficult to characterise by intact mass spectrometry. While a recent review by Sharma and colleagues notes the expanding role of mass spectrometry in the development of vaccines in general [15], others do not list MS as a potential technique for mRNA vaccine biochemical and biophysical product characterisation [16]. Nevertheless, Jiang and colleagues recently showed that mapping of the sequence of therapeutic mRNAs (~3000 nucleotides long) through LC-MS/MS is a viable option [17]. To accomplish this, they used parallel ribonuclease digestions, resulting in shorter oligonucleotides of different lengths, depending on the enzyme used. These techniques can be considered orthogonal to next generation sequencing (NGS) approaches and are able to detect single nucleotide polymorphisms (SNP) with a sensitivity of more than 99%.

In order to obtain or confirm the identity of an oligonucleotide, and by extension for all molecules, it is useful to compare the experimentally obtained isotope pattern of the molecule to the one that is theoretically expected based on its elemental composition [18,19,20]. While this is feasible when the identity of the molecule under investigation is known, it is hard to automate and becomes impossible if one wants to use the experimentally obtained isotope pattern to strengthen the identification of an unknown oligonucleotide. In peptide-centric MS-based proteomics, several algorithms have been developed that accurately predict isotope distributions based on observed mass values only, without the need for its elemental composition [21,22,23]. To our knowledge, comparable algorithms are not available for nucleic acid-based molecules, including (therapeutic) oligonucleotides. As mass spectrometry technology becomes more adopted for the analysis of nucleic acids, we foresee that bioinformatics tools that can forecast the expected isotope patterns generated by unknown DNA and RNA molecules will be of use for processing mass spectral data in order to reduce the dimensionality of the data and to help elucidate the elemental composition.

The current work describes a compositional data model that uses all possible in silico generated oligonucleotides up to a pre-specified length. The theoretical aggregated isotope distribution is then computed by the BRAIN algorithm [21,22] using the most recent NIST [24] definition for the elemental isotopes. The so-obtained molecular isotope distributions, which are compositional in nature, i.e., the isotope probabilities sum to one, are modelled with a statistically sound yet simple linear regression model for each isotope separately. The performance of the model is validated on a small set of DNA and RNA molecules that are repeatedly recorded on a Time-of-Flight instrument. We can already announce that our model’s approximation of the theoretical isotope distribution is justified because the error introduced by the average model is negligible in light of the error introduced by the instrument variability on the isotope intensities.

Because the modelling approach is kept simple, the prediction model is available as an online tool at https://valkenborg-lab.shinyapps.io/pointless4dna/ (version 1.00, accessed on 16 June 2021). The online tool provides the predicted isotope distribution for an average DNA or RNA molecule of a particular mass. The web interface presents the isotope information as a table and graphical representation. A batch version of the software is available as well. A user can upload a monoisotopic mass list as a text, csv or excel file, and the predicted isotope distributions can be downloaded as a formatted csv file. Furthermore, to facilitate the take up of this tool in a user’s workflow, an R-function that incorporates the parameter values of the polynomial model can also be downloaded from the website. In fact, users can implement these parameters in their favourite workflow manager or use a simple matrix multiplication and transformation to predict the average isotope distribution.

## 2. Material and Methods

### 2.1. Experimental Procedures

Oligonucleotides were synthesised via solid phase chemical synthesis at Janssen Pharmaceuticals (Beerse, Belgium) or obtained from Integrated DNA Technologies (Leuven, Belgium). More information about the DNA and RNA molecules can be found in Table 1.

The oligonucleotides for the proof-of-concept study were analysed via LC-MS by injecting 10 µL of a ca. 0.1 mg/mL aqueous solution on an H-Class UPLC (Waters Inc., Antwerp, Belgium) coupled to a Synapt G2 HDMS QTOF mass spectrometer (Waters Inc.). Chromatographic separations were performed using an Acquity BEH 300 C18 column (150 × 2.1 mm, 1.7 µm particle size) (Waters Inc.). The column heater was kept at 75 °C and the flow rate was 0.25 mL/min. mobile phase A consisting of 7 mM triethylamine and 60 mM 1,1,1,3,3,3-Hexafluoro-2-propanol (HFIP) in water, and mobile phase B was a methanol–acetonitrile mixture (50/50, *v*/*v*). The gradient elution consisted of a linear gradient of 0% to 70% of eluent B in 30 min. followed by a washing step of 5 min at 70% mobile phase B.

High resolution accurate mass data were acquired in negative ion mode using an electrospray ionisation source using a mass range of *m*/*z* 50 to *m*/*z* 2000 at a resolving power of approximately 12,000 (sensitivity mode, measured at *m*/*z* 1000). The following source conditions were applied: capillary voltage 2 kV, cone voltage 30 V, Extractor voltage 4 V, source temperature 120 °C, desolvation temperature 350 °C, cone gas flow 20 L/h.

The raw data was first converted to mzXML format using MSConvert GUI [25]. Converted LC-MS data files were read into R (v4.0.2) to be visualised and processed by the MSnbase package [26,27]. Chromatograms of each oligonucleotide strand were manually evaluated to determine their elution range (retention time range of the chromatogram peak) and detected charge states (Table 1). To generate isotopic envelope traces, mass scans in the retention time range were extracted and centroided, and mass peaks below the baseline intensity level of 50 counts were filtered. Isotopic envelope traces (two columns: mass, intensity) were extracted from selected mass scans according to predicted mass ranges. These mass ranges were calculated based on detected charge states and the monoisotopic molecular weight (MMW) of the oligonucleotide strand (between MMW and MMW + 15 Da). For instance, the *m*/*z* range extracted for charge state 6 of the DNA_SHORT1 strand is between 1386.6 (corresponding to MMW) and 1388.9 (MMW + 15 Da). For model validation, isotopic envelope traces of charge state 6 were evaluated in 10 LC-MS scans (in the elution range 10.95–11.05 min). Combining seven charge states, we obtained in total 70 replicates of isotopic envelop traces for evaluating isotopic distribution prediction of DNA_SHORT1.

### 2.2. Theoretical Data

A theoretical database of DNA oligonucleotides was created by generating all possible combinations of length 5 to 92, composed of the 4 DNA bases, Adenine, Cytosine, Guanine and Thymine with a 2′ deoxyribose phosphodiester backbone. Similarly, a database of RNA oligonucleotides was created containing all possible combinations of nucleotides Adenine, Cytosine, Guanine and Uracil with a ribose phosphodiester backbone from length 5 to 90 for RNA molecules. The elemental composition of the theoretical oligonucleotides is calculated using the information on the basis structures for nucleotides in Appendix A, further taking into account that for each added nucleotide, a molecule of water is lost. Consequently, for an oligo of length l, (l-1) water molecules are subtracted.

There are various bioinformatics tools that can calculate the theoretical isotope distribution, as described by Valkenborg et al. [19], but we prefer the BRAIN method to rapidly compute the aggregated isotope distribution, which also provides us with an exact computation of the centroid mass for each isotope variant. The elemental compositions are used as input for the BRAIN algorithm to determine the theoretical aggregated isotope distribution for each of the oligonucleotides. The first 20 aggregated isotope variants of the DNA and RNA compounds are computed using the most recent NIST definition for the elemental isotopes. Note that the choice of 20 isotopic variants is arbitrary, but as we discuss later in the manuscript, it covers up to 95% of the aggregated isotope distribution for the highest molecular weight DNA/RNA molecules. The goal of this work is to predict the expected isotope distribution of an average DNA/RNA molecule based on its monoisotopic mass using a simple regression modelling approach, which requires the data to be real-valued. However, the mathematical function used to calculate the theoretical isotope distribution is based on a multinomial expansion [28]. The probabilities of a multinomial distribution always sum to one, which can be referred to as compositional data. The isotope distribution that denotes the probability to observe a certain isotope variant of a molecule can, in theory, be viewed as compositional data; however, it does not follow a classical multinomial distribution. The number of isotopes and the distribution of the probabilities vary in function of the mass. For the low mass compounds in our database, the first eight aggregated isotopes cover 100% of the isotope probabilities, whereas for some high mass compounds the entire isotope distribution is not covered by the first 20 isotopic variants. Since we aim to produce a model that is applicable for the entire mass range for oligonucleotides of length 5 to 92 and length 5 to 90 for DNA and RNA molecules, respectively, we need to manipulate our data to become truly compositional. One solution is to include more isotope variants in the model such that the isotope probabilities always sum to one. However, this would lead to many isotope components with almost zero probability in the lower mass range, which compromises the regression modelling approach. Instead, it is solved by introducing a sink-hole that contracts the leftover probabilities into one isotope variant, called the pseudo-isotope, or, more correctly, in compositional data jargon the closure term. By introducing this pseudo-isotope, we ensure that 100% of the isotope information is captured by the first 20 isotopic components and the additional closure component.

### 2.3. Compositional Data Transformation

The elements of our composition, i.e., the isotope probabilities, are nonnegative and constrained to sum to one by the creation of the pseudo-isotope. From the work of Aitchison [29,30,31,32] it is known that compositional data such as the calculated isotope distribution can be represented on a high-dimensional simplex. Consider the vector $x=\left(x_{1},\;x_{2},\;\ldots \;,\;x_{D}\right)$ to be a *D*-part composition of proportions $x_{i}\;\left(i=1,\ldots ,D\right)$. The composition satisfies the unit-sum constraint $x_{1}+x_{2}+\ldots +x_{D}=1$, thus the effective dimension of the *D*-part composition is reduced to d=D−1, where:(1)$$x_{D}=1-\overset{d}{\underset{i=1}{\sum }}x_{i}$$

Therefore, the compositional sample space can be accurately represented by the *d*-dimensional unit simplex:(2)Sd={x1,…,xD ϵ ℝD |xi≥0 for i=1,…,D  and ∑i=1Dxi=1}

The compositional data structure imposes constraints on a regression modelling approach since the sample space for our data is restricted to the unit simplex $S^{d\;}$, as opposed to the entire real space $R^{D}$. To remediate these constraints and facilitate regression, the compositional data is transformed before analysis. A composition only provides information on relative and not absolute values, so any relevant function of the components must be expressible in terms of ratios of the components. Therefore, the Aitchison simplex can be transformed into real space via some well-characterised isomorphisms [32]. These include the following transformations: the additive log-ratio transformation (ALR), the center log-ratio transformation (CLR) and the isometric log-ratio transformation (ILR). In case of the CLR transformation, a notion of a geometric mean is needed, which is not trivial in our case. Moreover, the CLR transformation forces the component covariance matrix to be singular. To apply the ILR transformation, a notion of central tendency is needed as well as complex calculations. Therefore, we have opted for the ALR transformation which can easily be applied even if the isotope distribution is only partially observed, provided that the reference probability is detected. Its simplicity allows us to make easy predictions using the regression coefficients, and the presence of a clear reference component allows us to represent the observed spectrum in reduced ALR space as further discussed in the goodness-of-fit section. Furthermore, it should be remarked that the ALR transformation is not an isometry; therefore, distances in the ALR space are not retained. This could be solved by employing the ILR transformation rather than the ALR. However, another disadvantage of the CLR and ILR transformations is that they both require the entire isotope distribution to be seen that span further than the first 20 peaks in our case. The ALR transformation is the only transformation that can handle incomplete observations of the isotopic envelope and can handle a partial distribution to perform the back-transformation from real space to the Aitchison compositional unit simplex, provided that the reference peak is observed.

Let $x=\left(x_{1},x_{2},\ldots ,x_{21}\right)$ denote the vector representing the first 20 aggregated isotope probabilities and the pseudo-isotope, so 21 elements in total. Then, the ALR transformation is given by:(3)$$S^{d}\;\to \;R^{d}:\;\left(x_{1},x_{2},\ldots ,x_{21}\right)\;\to \;\left(ln\left(\frac{x_{2}}{x_{1}}\right),\;ln\left(\frac{x_{3}}{x_{1}}\right),\;\ldots \;,\;ln\left(\frac{x_{21}}{x_{1}}\right)\right)=\left(z_{1},z_{2},\ldots ,z_{20}\right)$$
With *d =* 20. Note that the monoisotopic peak ($x_{1}$) is chosen as the reference component. Since standard multivariate statistical methods are invariant to the choice of reference category, the choice of divisor in the transformation in principle does not influence the results or the interpretation [30,33]. Note that, in this manuscript, it is convenient to fix the reference for the entire mass range to obtain a general model for the envisioned mass range. We opt for the monoisotopic variant as a reference because this variant is dominantly present for low molecular weight oligonucleotides. However, a user should be aware that the probability of the monoisotopic variant is a function of the molecular weight of the DNA or RNA molecule; therefore, the detection or signal-to-noise ratio of that monoisotopic variant is not uniform over the mass range. To remediate this drawback, a different approach can be considered that divides the envisioned mass range in distinct regions and constructs a separate polynomial model with the optimal choice for the reference for each region. In every region, the optimal choice would be the variant that is most likely above the detection limit and with the maximum signal-to-noise ratio.

### 2.4. Modelling Approach

We adopt a machine learning approach where a univariate polynomial regression model is trained to model the underlying relation between the monoisotopic mass of a DNA or RNA molecule and its corresponding isotopic envelope (containing the first 20 isotopes and pseudo-isotope), transformed into ALR space. Before performing the regression analysis, a randomly selected test set containing 5% of the observations is left out. A univariate weighted least squares polynomial regression model, using the squared residuals of the ordinary least squares model as weights, is fitted by minimising the sum of squared errors on each ALR transformed isotope separately. Forward model selection to determine the optimal order of the polynomial is performed using the mean squared error (MSE) on the test set to assess model performance. The final model is then trained on the entire dataset (training and test set combined). Let the monoisotopic mass of a DNA/RNA molecule be denoted by *m*, then the resulting polynomial models of order *k* are given by
(4)$$z_{1,i\;}=\beta_{1,0\;}+\beta_{1,1\;}m_{i\;}+\beta_{1,2\;}m_{i}{}^{2\;}+\ldots +\beta_{1,k\;}m_{i}{}^{k\;}+\varepsilon_{1,i}\;z_{2,i\;}=\beta_{2,0\;}+\beta_{2,1\;}m_{i\;}+\beta_{2,2\;}m_{i}{}^{2\;}+\ldots +\beta_{2,k\;}m_{i}{}^{k\;}+\varepsilon_{2,i}\;\ldots \;z_{20,i\;}=\beta_{20,0\;}+\beta_{20,1\;}m_{i\;}+\beta_{20,2\;}m_{i}{}^{2\;}+\ldots +\beta_{20,k\;}m_{i}{}^{k\;}+\varepsilon_{20,i}\;with\;\varepsilon_{j,i}\sim N\left(0,\sigma_{j}^{2}\right)\;for\;j\;\in \left(1,\ldots ,20\right)$$

Apart from a model to predict the probabilities, a method is provided to predict the centroid mass for the corresponding isotope variants. This method simply computes the average mass differences between the aggregated isotope variants and the monoisotopic variant for all the molecules in the DNA and RNA database separately. The result of this calculation is a simple vector of mass differences to which a monoisotopic mass is added.

### 2.5. Prediction of the Isotopic Envelope

The result of the fitting procedure yields a $20\times \left(k+1\right)\;$matrix containing the estimated parameters for the intercept and *k* polynomial coefficients, for each of the 20 ALR transformed isotopes. As such, the obtained model can be used to predict the expected ALR-transformed isotopes $\left(z_{1},\ldots ,z_{20}\right)$ based on the monoisotopic mass *m* via simple matrix multiplication.
(5)$$\left[\begin{array}{c} z_{1}\\ \begin{array}{c} z_{2}\\ \vdots \end{array}\\ z_{20} \end{array}\right]=\left[\begin{array}{c} \beta_{1,0}\\ \begin{array}{c} \beta_{2,0}\\ \vdots \end{array}\\ \beta_{20,0} \end{array}\begin{array}{c} \beta_{1,1}\\ \begin{array}{c} \beta_{2,1}\\ \vdots \end{array}\\ \;\;\;\beta_{20,1} \end{array}\begin{array}{c} \cdots \\ \begin{array}{c} \cdots \\ \vdots \end{array}\\ \cdots \end{array}\begin{array}{c} \beta_{1,k}\\ \begin{array}{c} \beta_{2,k}\\ \vdots \end{array}\\ \beta_{20,k} \end{array}\right]\cdot \left[\begin{array}{c} 1\\ \begin{array}{c} m\\ \vdots \end{array}\\ m^{k} \end{array}\right]$$

To obtain the isotopic probabilities, these ratios need to be back-transformed to the original simplex space. This is accomplished by using a modified softmax transformation, which is a well-known transformation in the field of artificial neural networks. By adding an extra column of ones before employing the softmax transformation, we control the probabilities predicted by the unconstrained linear polynomial regression model to produce estimates that sum back to one.
(6)ℝd→Sd:z1,z2,…,z20→x1,x2,…,x21=1gz,ez1gz,…,ez20gzwith gz=1+ez1+…+ez20
with *d =* 20 or giving *D* = 21 predictions for the first 20 isotopic variants and the closure term, or, equivalently, the pseudo-isotope.

### 2.6. The Goodness-of-Fit Statistic

The performance of the theoretical model is tested on the benchmark dataset described in the experimental procedures section. The theoretical model predicts the first 20 isotopes both as ALR transformed isotopes and as probabilities back-transformed via the modified softmax function. We consider a goodness-of-fit statistic in the ALR space that is inspired by the metric used to fit the univariate model and a metric in the simplex space (i.e., probabilities) that is inspired by the multinomial test. In a real-life dataset, not all 20 isotopic variants are always seen. Moreover, in our validation dataset, the MMW to MMW +15 Da range was extracted, thus we expect approximately 15 isotopes. Note that on some occasions, the monoisotopic variant could not be retrieved from the data. Since we know the underlying compound, we know exactly which peaks have been detected. Therefore, we opt to work with metrics that are applicable on an only partially observed isotope cluster.

Let $\left(o_{1},\ldots ,o_{k}\right)$ be the observed isotope pattern, with $o_{1}$ representing the monoisotopic peak and $o_{k}$ the last observed isotope variant. To calculate the ALR goodness-of-fit metric, the observed spectrum is first transformed to ALR space by applying the log-ratio transformation using the monoisotopic peak as reference, such that:(7)$$\left(t_{1},\ldots ,t_{k-1}\right)=\left(\ln\left(\frac{o_{2}}{o_{1}}\right),\ldots ,\ln\left(\frac{o_{k}}{o_{1}}\right)\right)$$

The predicted ALR ratios $\left(z_{1},\ldots ,z_{k-1}\right)$ are computed using the separate univariate linear regression models described in the previous section. Consequently, the transformed observed isotope ratios are compared directly with the predicted ratios in Aitchison ALR space by computing the squared errors. Finally, we calculate the mean squared error across isotopes for each molecule to assess the fit of the model and thus:(8)$$MSE_{ALR}=\frac{1}{k-1}\overset{k-1}{\underset{i=1}{\sum }}{\left(t_{i}-z_{i}\right)}^{2}$$

Comparing ratios directly in the ALR space is very convenient since it does not call for scaling the intensities. However, the requirement for this metric to work is that the monoisotopic variant is observed since it serves as a reference in our modelling approach.

Let $\left(o_{l},\ldots ,o_{k}\right)$ be the observed isotope pattern, with $o_{l}$ and $o_{k}$ representing the first and last observed isotopic peak variant, respectively. To calculate the goodness-of-fit in the simplex (i.e., probabilities) space, we adopt an error statistic that is inspired by (but by no means equal to) the multinomial test introduced by Pearson [34]. The error is computed as:(9)$$\chi_{\mathrm{simplex}}^{2}=\frac{1}{k-l+1}\overset{k}{\underset{i=l}{\sum }}\frac{{\left(O_{i}-E_{i}\right)}^{2}}{E_{i}}$$
where $E_{i}=Nx_{i}$ is the expected peak intensity, with N being the total peak intensity and $x_{i}$ the predicted probabilities using the theoretical models, back-transformed using the modified softmax transformation. The theoretical model is built using the aggregated isotope distribution with probabilities that sum to one. Since N is unknown when not all isotopes are detected in the spectrum, N is calculated as the ratio of the sum of observed intensities and the sum of the probabilities of the expected peaks.
(10)$$N=\frac{\sum_{i=l}^{k}O_{i}}{\sum_{i=l}^{k}x_{i}}$$

Note that the values *l* and *k* change with each compound and, therefore, the degrees of freedom change with each spectrum. For this metric, while it is essential to know which peaks have been observed in the spectrum, it is not necessarily required to observe the monoisotopic variant. However, knowledge of the monoisotopic mass is necessary to use the prediction model.

## 3. Results

### 3.1. Generation of the Data Sets

The DNA and RNA datasets span a length from 5 to 92 nucleotides and 5 to 90 nucleotides, respectively, spanning a mass range from 1463.2424 Da to 30,290.8424 Da and 1543.2170 Da to 31,072.2797 Da. A total of 3,321,890 and 3,049,431 possible nucleotide combinations (not permutations) are considered for DNA and RNA. The lightest oligo is composed out of 5 times dCMP or 5 times CMP, and the heaviest oligo is composed out of 92 times dGMP or 90 times GMP for DNA and RNA, respectively. It is worthwhile to mention that the DNA and RNA nucleotide combinations all have a unique elemental composition and, consequently, a unique mass. Such behaviour cannot be observed for amino acids, where different amino acid combinations can boil down to exactly the same elemental composition and mass.

The histogram in Figure 1 displays the abundance of the different oligonucleotides over the mass range. Two observations can be made from this plot. Firstly, note that the density of the data increases exponentially as a function of the mass Secondly, it should be noted that the dataset is no longer representative at the high mass range, indicated by the sharp drop in density at the end of that mass range. This artefact is caused by how the theoretical oligonucleotide database is created. Our approach enumerates all possible nucleotide combinations up to a pre-specified length. Thus, the lightest possible DNA oligo of length 93, that is not a part of the database, would be purely composed out of 93 dCMPs, which yields a mass of 26,899.3232 Da. This mass is indicated by a vertical red line in Figure 1. Starting from this mass, the data is not representative as some nucleotide variants of larger length are depleted from the data. In case of the RNA database, this limit would be a molecule that contains 91 CMPs with a mass of 27,776.7677 Da. For this reason, we argue to restrict the prediction model to the highest mass in the database just below the aforementioned limits, or, equivalently, to the range [1463.2424–26,899.3222] Da containing 2,631,058 DNA molecules and [1543.2170–27,776.7667] Da containing 2,557,189 RNA molecules. A restriction for the lower mass is not needed as the highest mass molecule of length four remains far away from this lower limit.

The scatter plot in Figure 2 displays the first 20 isotopes of all possible DNA molecules within the restricted mass range. The *x*-axis indicates the monoisotopic mass of the oligonucleotides. Every aggregated isotope variant is denoted with a different colour, and their probability is provided on the *y*-axis. The fast declining isotope in blue on the left-hand side of the plot is the monoisotopic variant. It can be observed that the probability of this variant is decreasing fast, which makes this variant often fall below the instrument’s detection limits for DNA molecules of a molecular weight larger than 10,000 Da. The black line on the top of the figure is the coverage which sums the probabilities of the first 20 isotopes. For low mass DNA molecules, a coverage of 100% is reached, but this is not the case for high mass molecules, where coverage drops towards approximately 95%. This coverage is used to compute the closure term (i.e., the pseudo-isotope) by subtracting the coverage from 100%.

The flexibility of the prediction model, i.e., the order of the polynomial, is chosen by minimising the MSE on a test data set. As the objective is to obtain an optimal prediction within the operational mass range of our model, a random selection of oligonucleotides is taken from the restricted mass region mentioned before. We choose a test set of approximately 5% of the oligonucleotides in the data base, resulting in 166,095 and 152,472 instances with a mass range of [1493.2418–26,899.3048] Da and [1545.1850–27,776.7557] Da for DNA and RNA molecules, respectively. We argue that such a validation set approach (opposed to k-fold or leave-one-out-cross validation) is sufficient and representative given the large data volumes. As training data, we use the remaining molecules present in our database that span the entire mass range. The motive for this counterintuitive choice is to remediate potential disadvantageous effects at the high-mass boundary caused by the polynomial model.

### 3.2. Model Selection and Training of the Model

This section describes the selection of the polynomial order of our prediction model and the final model fit. The description is confined to the DNA data as the procedure for the RNA model is exactly the same. The graphics for the RNA model performance can be found in the online Appendix A. Figure 3 shows the scatterplot that depicts the relation between the ALR transformed isotopes, using the monoisotopic variant as reference, and the monoisotopic mass. The ratio derived based on the closure term is denoted by the ALR20 isotope.

Every ALR transformed isotope is modelled separately by a univariate polynomial model. We consider polynomials of order 1 to 15. To keep the polynomial regression well-conditioned, we standardise the monoisotopic mass covariate in our model by subtracting the mean (mu = 22,746.1953) of all the mass values in our database and dividing the difference by the standard deviation (sigma = 4788.8776). Appendix A illustrate how the test MSE of each model evolves as a function of the order of the polynomial. From the sixth order onwards, the test MSE improves only very slightly. The majority of the models reach their minimum value at a polynomial order of 10, while for higher orders the MSE slightly increases and becomes unstable. For this reason, we select a polynomial of order 10 as our final model and retrain the model on the complete, unrestricted dataset. This results in a 20 × 11 matrix containing the estimated parameters for the intercept and 10 polynomial coefficients, for each of the 20 ALR transformed isotopes. The model parameters can be downloaded from the online tool as an R function. The white line in Figure 3 plots the predicted values of the ALR-transformed isotopes. After back-transforming the ALR isotopes via the modified softmax function, the predicted isotope probabilities are obtained. These predicted probabilities for the first 20 aggregated isotopes of our DNA and the coverage term are indicated in Figure 2 by a white line. For both figures it is clear that the predicted values are at the centre of the data cloud. A proper fit can also be witnessed by the residual plot in Appendix A. For the majority of the isotopes the residuals remain within an error interval between 0.004 and −0.006. These numbers demonstrate that the approximation of the theoretical isotope distribution by an average model is justified as it only introduces a small deviation.

In order to better understand the errors introduced in the next section, we provide the mean squared errors between the theoretical ALR transformed isotopes and the predictions from the model on one hand, and the mean Pearson’s chi-squared error between the theoretical probability values and the softmax transformed predicted probabilities on the other in Figure 4.

From Figure 4a, it can be observed that the low mass region has the highest error. There are two explanations for this behaviour. Firstly, the boundary effect of the polynomial could play a role. However, a second, more probable explanation is that the data in this low mass region is too sparsely distributed in contrast to the high mass region such that an imprecise model fit is less penalised. This effect is also the reason for choosing the weighted regression approach, where weights are chosen as the residuals of the ordinary least square fit. In general, the error is small when contrasted to experimental data as shown in the next section. The pattern in Figure 4b contrasts this observation. As the mass increases, the probabilities are spread over more peaks, which leads to a larger error, reflecting the increase in degrees of freedom of the multinomial distribution. The decline of the Pearson’s chi-squared error from 20 kDa is caused by an increase of the pseudo-isotope probability, which is not part of the error calculation as presented in Equation (9). Note that although the mean value is taken, all DNA molecules in the theoretical database have the first 20 aggregated isotope variants returned, even if they are near zero values. Nevertheless, Figure 4 provides an informative visual interpretation of the variability that we can expect for the errors across the mass range. Appendix A presents a similar plot for the RNA prediction model. Similar conclusions can be drawn from the RNA model, with the addition that a slight bias is introduced at the low masses starting at 1545 Da and around a mass of 4000 Da. It seems that the polynomial model of order 10 lacks flexibility to correctly capture the isotope trends as seen in Appendix A.

Besides the probabilities, the model also provides a prediction for the centroid mass of the aggregated isotope variants. This prediction is simply taken as the average value of the mass differences between the isotope variant and the monoisotopic variant across the molecules in the DNA and RNA database. The results of these average mass differences are listed in Table 2. There is only a minor difference between the DNA and RNA molecules. Nevertheless, we use these values to compute the centroid masses of the predicted isotope probabilities by simply adding these values to the observed mass of the monoisotopic variant. Although there is a relation between the residual error of the mass model and the actual centroid mass of the molecule, as illustrated in Appendix A, we will neglect this trend as the maximum absolute error is below 6 mDa.

The same methodology is followed to model the data from the RNA database of oligonucleotides. A polynomial order of 10 is found to be optimal in terms of minimising the test MSE. The Appendix A display all the data (Appendix A) and plots (Appendix A) for RNA that also have been discussed in the context of the DNA model.

### 3.3. Exploration of the Proposed Error Metrics

The final polynomial model is tested on the experimental spectral data of DNA and RNA oligonucleotides described in the material and methods section. As the nucleotide composition of these compounds is known, we first compare the experimental data with the theoretical isotope distribution computed by BRAIN. In this analysis, we investigate whether our proposed metrics depend on the intensity or charge state and evaluate whether the assumption on the elemental isotope definition provided by NIST is compatible with laboratory-grade synthesised DNA/RNA strands. Next, we move the discussion to the comparison of the theoretical data with the predictions from the DNA and the RNA model, to investigate whether the error introduced by the average model is acceptable compared to the error introduced by the instrument variability. In this discussion, we limit ourselves to only two of the four DNA strands, namely, DNA_short1 around 8325.41 Da with high quality isotope patterns, and DNA_long around 14,426.37 Da with lower quality isotope patterns. The results for the other two strands are presented in the Appendix A.

We start our investigation with compound DNA_short1 that is found in the spectral data at charge states 6 up to 12. Around the apex of the chromatographic elution profile, 10 spectra are selected for investigation. As a result, we have 70 observed isotope patterns for this DNA molecule originating from different charge states, with varying intensities, and consequently with a varying number of detected isotope peaks. Two isotope clusters were discarded from this dataset because the monoisotopic peak could not be retrieved. Figure 5a displays the boxplots of the error in the ALR space across the different charge states. Appendix A Appendix A showcase the distribution of the number of isotope peaks and the sum intensity of the observed isotope cluster as a function of the same charge states. It can be seen that the intensity and the number of detected peaks increase with charge state, suggesting that charge 12 is the dominant isotope cluster in the spectrum. Although the number of observed peaks and the intensity values severely fluctuate within and across the charge states, the MSE in the ALR space is stable and comparable across the different charges. Such an intensity and peak number invariant score is especially convenient when a global threshold is applied to remove low quality fits, as this threshold can be uniformly applied for all DNA and RNA oligonucleotides, disregarding the ion statistics. On the other hand, when looking at the boxplots in Figure 5b, we see that the mean Pearson’s chi-squared error (MPCSE) increases with increasing charge state. However, from Appendix A it becomes clear that the number of detected isotope peaks increases with charge state, and consequently, the sum intensity increases as well. The calculation of the MPCSE as presented in Equations (9) and (10) takes into account the total sum intensity of the observed isotope distribution in the spectrum. Therefore, the underlying pattern in Figure 5 can be explained by the intensity dependency of the error metric. This property is convenient when quality thresholding needs to take into consideration the ion statistics of the isotope pattern. For example, a high total ion count, which means that there is sufficient sampling of the multinomial isotope distribution, should yield a small relative error, i.e., proportional to the height of the isotope peak. The opposite is true for low abundant spectra, where we expect a larger relative error because of the low number of ions sampling the distribution. The MPCSE will compensate for this effect by the term in the denominator that normalises the mean squared error by the expected intensity. This property, however, ignores some important nuisance effects that can come into play with time-of-flight mass spectrometry, one of which is the detector saturation for high intense signals. Saturation causes flattening out of the high peak intensities leading to a larger error between the observed and theoretical isotope distributions, an error that cannot be tolerated given the high ion statistics.

In order to combine the benefits of both types of scores, a new graph is proposed, shown in Figure 6. Each point represents one isotope pattern of compound DNA_short1 and its corresponding scores: the MSE in the ALR space on the *x*-axis and the MPCSE in the intensity space on the *y*-axis. The log10 of the sum intensity is coded as a third colour dimension. Some interesting conclusions can be drawn from this new visualisation. An isotope cluster, indicated by circle (a) in Figure 6 and the theoretical and observed isotope pattern in Figure 7a, illustrates that the observed and the theoretical isotope distributions are in agreement in the ALR space because of the low MSE of 0.2827. However, the MPCSE is very large with a value of 229.85. This is caused by the high total intensity of this isotope cluster as is illustrated by the yellow colour in Figure 6 and the sum intensity in the title of the Figure 7a. A possible explanation could be very mild saturation of the detector leading to a deviation that is not tolerable in view of the high ion statistics. The opposite can be observed for another isotope cluster denoted by circle (b) in Figure 6 for which the isotope patterns are visualized in Figure 7b. The ALR score is worse in this case, reaching a value of 1.3, but it can be seen that the MPCSE is lower than in Figure 7a whilst the fit is visually less consistent. This behaviour can be explained by the lower summed intensity value of 25,635 that better tolerates a discrepancy in the goodness-of-fit. Data points (c) and (d) are presented in Appendix A Appendix A and denote spectra with low ion statistics. We recommend the users of this model to always conduct an experiment of a DNA/RNA standard to benchmark and calibrate the method for your laboratory set-up. By doing so, you will understand the normal range of the proposed error statistic and can argue on a thresholding criterion that is tailored for your purposes.

### 3.4. Model Validation

Next, we investigate whether the prediction from the average isotope model contributes substantially to the goodness-of-fit error compared to the error between the observed and the theoretical isotope distribution. Figure 8 presents three boxplots that showcase the distribution of the MSE in the ALR space for the 68 observed isotope patterns of compound DNA_short1. Interestingly, the MSE in the ALR space for the predicted isotope distribution via the DNA model (middle boxplot) is similar, though slightly higher than for the theoretical model (left boxplot). This result is logical as the DNA model will predict the isotope distribution for an average DNA oligonucleotide of that particular mass. Compared to the theoretical model, this introduces a bias, but from Figure 8 it can be seen that this bias only has a minor contribution with regard to the total error. The third box in Figure 8 is the error when using the RNA prediction model. Note that the errors here are severely inflated due to the use of a misspecified model, i.e., comparing a predicted RNA isotope distribution with the observed isotope distribution from a DNA oligonucleotide. A valuable conclusion here is that one can discern a DNA oligonucleotide from an RNA oligonucleotide in a label-free manner just by investigating the resulting isotope profile in a mass spectrum.

Another valuable remark here is that the boxplots presented in Figure 8 might be slightly misleading. The boxplot representation obscures the fact that the scores between the boxes are correlated since the computed errors in each box are based on the same 68 observed isotope distributions. A plot that makes this point more clear is displayed in Appendix A Appendix A. This spaghetti plot represents each of the 68 molecules by a line. The lines between the errors from the theoretical and DNA prediction model are near horizontal, indicating that it does not matter substantially whether you would have the exact theoretical isotope distribution or the predicted average isotope distribution for pattern recognition and spectral processing. In contrast, the lines between the errors for the DNA and RNA prediction model illustrate a clear incline, indicating that model misspecification does matter. Another way of quantifying the effect of model misspecification is to use the Wilcoxon signed-rank test (i.e., non-parametric paired t-test) in order to assess whether the error differences between the three models are significant. The result of this test for theoretical versus average DNA gives a *p*-value of 4.4844 ×10^−7^ with an effect size of 0.0136, whereas the comparison of the DNA model with RNA results in a *p*-value of 2.0060 × 10^−10^ that is even more significant and has a higher effect size of 0.1221. The latter principle can be adapted for the inclusion in automated processing workflow as label-free DNA/RNA classifier. Further, it is worthwhile to emphasise here that although the error between the theoretical and DNA prediction model is statistically significant, it is not relevant for our application given the small effect size that falls within the variability of the instrument. This can be seen in Figure 4a, where the effect size of 0.0136 falls within the expected error range, whereas the RNA model is out of specification with an error of 0.1221.

Next, we discuss the results of the second compound DNA_long. Since this compound has produced lower quality spectra with many peaks missing, we have opted to compute an average observed isotope distribution across the 23 scans for each of the four charge states. Figure 9 provides the averaged observed isotope distribution rescaled towards probabilities (blue stems), along with the predicted isotope distributions from the average DNA model (red stems) for the charge states ranging from z = 15 (panel a) up to z = 18 (panel d). The MSE and MPCSE in the titles of the figures indicate that this goodness-of-fit is in range with previous observations, proving that the prediction model is also suited for larger DNA molecules.

Lastly, Appendix A Appendix A contains the boxplot of the ALR error of the RNA-like molecule. There is a large difference in error when comparing the observed isotope pattern with the theoretical one or with the predicted isotope distribution from the DNA model. Such a discrepancy is expected due to the model misspecification. However, the error for the predicted RNA model is even larger. This observation warrants further investigation, but we argue that the large error here is also caused by model misspecification since the molecule under scrutiny is composed of modified RNA that contains sulphur and fluorine and our DNA/RNA model is not capable of accommodating these elements. Having said that, we can also conclude that, even for a misspecified model, the ALR error is mostly below one, which is acceptable when compared to the other compounds in the dataset.

### 3.5. Software

The modelling approach described in Materials and Methods, with the polynomial order set to 10, has been implemented in Shiny (a package in R programming language) and deployed at https://valkenborg-lab.shinyapps.io/pointless4dna/ (version 1.00, accessed on 16 June 2021). The user interface is presented in the screenshot in Appendix A.

Firstly, the user has to choose the input type (single or multiple masses) and provide the corresponding input data accordingly in the text field right there in the app or by loading a text, csv or excel file with multiple masses for batch processing. The input file for batch processing should contain only one column (without any column header) with numerical values (dot as a decimal separator) representing the masses of interest. Note that for any specified mass value that exceeds the mass range used in the model fitting, the isotope distribution will not be computed and warning messages will be shown. In the following step, the molecule type has to be selected between DNA or RNA.

Once the input and molecule type is specified, the ‘Calculate’ button, which triggers the prediction of the isotopic envelope, appears in the user interface. After clicking the button, a zip file that contains csv tables with masses and predicted probabilities of isotopic peaks is offered for download via the ‘Download isotope distribution table’ button. In addition, if the user decides on the single mass input type, the predicted probabilities will be directly displayed in the app as a table and graph.

Lastly, the ‘Download R function’ button is available throughout the entire user session allowing the download of a zip file. This zip file includes the underlying R function to predict the isotope distribution and two. Rds files (one for DNA and one for RNA) storing information on the estimated polynomial coefficients in a 11 × 20 table, the mass range used in the model fitting, and the mean and standard deviation values applied when standardizing the mass covariate. This effortless access to the app’s components presents the user with the opportunity to quickly reuse our modelling approach in their own analytical pipeline.

## 4. Discussion

Since there is already much discussion in the results section, we limit the discussion to a bulleted list of additional remarks and reiterate some important characteristics of the prediction model and the associated goodness-of-fit statistics:Model fit: A polynomial model of order 10, which minimises the MSE on our test set, is acceptable given the vast amount of data points, making the risk of overfitting minimal. However, a few remarks are worth considering. Figure 2 illustrates that the density of data points increases with molecular weight. Since it is beneficial for a regression model to spend its flexibility in the dense data regions to minimise the MSE over the entire dataset, less flexibility remains for modelling the lower mass region. This effect is partly remediated by using a weighted least square regression approach that employs squared residuals as the weight. However, even these weighted errors do not contribute much to the overall MSE, leading to a biased model fit for the RNA model, as can be seen in panel (b) of the Appendix A. Furthermore, polynomial models with a high order are known to have irregular behaviour at the boundaries. In order to rectify this effect, we have used the full range dataset but restricted the prediction model to the range specified earlier. In future work, we will propose a model that is capable of handling these boundary effects and that can spread its flexibility more evenly over the data range even though the data is distributed in an unbalanced manner.Transformation of the simplex: The additive log-ratio transformation is the obvious choice for our modelling exercise since it can handle partially observed data. Here we opted for the monoisotopic variant as a reference. From Figure 2 it can be observed that the probability of this variant decreases rapidly with increasing molecular weight. For the current mass range, the division by small numbers remains within machine precision, but it can be argued that, for higher molecular weights above 30 kDa, a different reference has to be used to keep the transformation well-conditioned.Transformation of the observed isotope distribution: The benefit of the ALR transformation is that we can transform the partially observed data into ALR space and execute the goodness-of-fit comparison directly with the predicted ALR transformed isotopes. This strategy is only possible if the monoisotopic variant has a quantifiable intensity in an observed spectrum. Two remarks are worth considering. Firstly, as mentioned in the previous bullet point, the monoisotopic variant falls below the limit of detection for large molecules, obstructing the transformation of the observed isotope distribution in the ALR space. Secondly, a large error on the monoisotopic intensity will propagate severely in the ALR error as this reference is used to transform all the isotopes. To remediate this effect, we could argue to divide the mass range into distinct bins for which an optimal reference is chosen.Multinomial test: In order to relax the dependency of the goodness-of-fit statistic on the monoisotopic variant, we proposed a score that uses the back-transformed probabilities. This score can be computed on a partially observed isotope distribution, provided that the monoisotopic mass is known and the user has knowledge of which isotopes are observed. Another remark is that the multinomial test is approximated by the Pearson’s chi-squared test. However, our implementation does not rely on formal statistics. Furthermore, instead of the actual number of ions, the intensity values are used as a proxy. Therefore, the proposed mean Pearson’s chi-squared error just tries to quantify the goodness-of-fit. As discussed earlier, this metric is able to tolerate larger error deviations when intensities are low (i.e., low ion statistics) or can be more stringent for high intensities (i.e., high ion statistics). The latter states that a one-to-one relation between intensity and ion statistics exists, but the kind of relation depends on the instrument type. Hence, without this formal testing framework, the MPCSE is not interoperable between platforms. Therefore, we suggest calibrating this metric on every instrument using a benchmark dataset for optimal thresh old selection.Choice of the model covariate: The monoisotopic mass is used as the predictor variable in our model (*x*-axis in Figure 2 and Figure 3). Hence, to forecast the isotope distribution we need to have the monoisotopic mass as an input. From earlier discussion, we know that the monoisotopic variant is not observed for high molecular weight molecules. Three procedures can be proposed to solve this conundrum. Firstly, the same model can be devised but with a different covariate, for example, the average or most abundant mass of the molecules. Secondly, a model similar in spirit as MIND by Lermyte et al. [35] can be proposed for DNA that predicts the monoisotopic mass based on the partially observed isotope distribution. Thirdly, we can use the model in combination with the fitting procedure of Senko et al. [36] that scales the elemental composition of an average amino acid, denoted as averagine. However, for the use case in nucleic acids, this would require the counterpart of the averagine model for DNA and RNA.Univariate approach: From the residual plot on the predicted probabilities and our ad hoc method to compute the centroid masses, we can observe different types of correlation in the data. It is worthwhile to investigate whether this correlation can be exploited in a multivariate analysis in order to obtain a better prediction model. On the other hand, we have demonstrated that current predictions of probability and mass are very close to the actual values, and that the errors are ignorable given current instrument precision and mass accuracy.

## 5. Concluding Remark

In this manuscript, we have proposed a model that predicts the aggregated isotope distribution of an average DNA or RNA molecule given the monoisotopic mass as an input variable. The prediction model can be used to detect and deconvolute nucleic acid isotope patterns in a mass spectrum for which the elemental composition is unknown. To compare the predicted and observed isotope pattern, we have proposed two different goodness-of-fit statistics. The first statistic operates directly in the Aitchison geometry and is invariant for the observed intensities, but only works for smaller molecules for which the monoisotopic peak is observed in a spectrum. The second statistic is based on the multinomial test and is able to operate on a partially observed isotope distribution where the monoisotopic variant is not observed. This statistic is influenced by the intensity values and therefore needs proper calibration on a benchmark dataset.

The prediction model is evaluated on a dataset containing repeated measurements of four different DNA/RNA molecules. It can be concluded that the predictions made by the model are very close to the actual probabilities and mass values, and that the error can be ignored given the instrument variability.

The prediction model is not demanding in computational resources as it only requires matrix multiplication and simple back-transformation. The model is implemented as an online tool and can be downloaded from the website as an R function.

## Figures and Tables

**Figure 1 metabolites-11-00400-f001:**
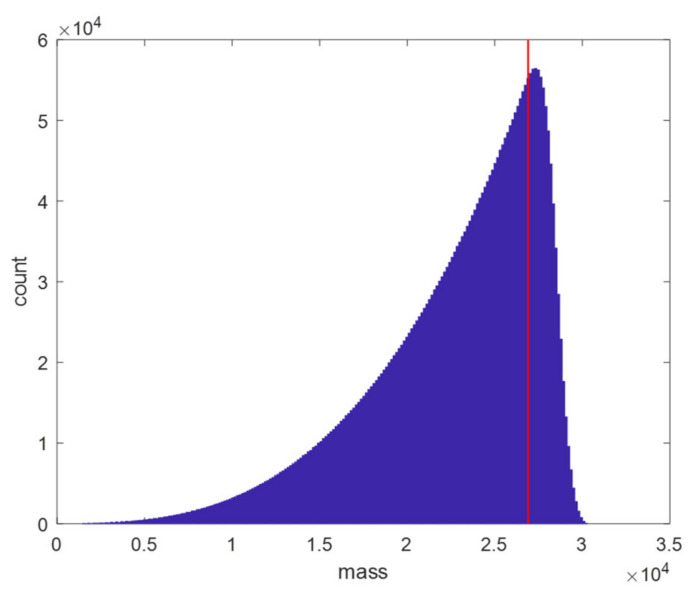
Histogram of the calculated mass of DNA oligonucleotides. The DNA molecules span a sequence length from 5 to 92. The number of possible nucleotide combinations (not permutations) grows exponentially with the mass of the DNA molecule. Only a few thousand oligonucleotides are situated at the lower mass range, while tens of thousands of oligonucleotides can be found in the higher mass ranges. The vertical red line indicates the lightest DNA sequence of length 93 composed out of dCMP only. From that mass on the data set is not representative anymore since longer DNA molecules are depleted from the dataset.

**Figure 2 metabolites-11-00400-f002:**
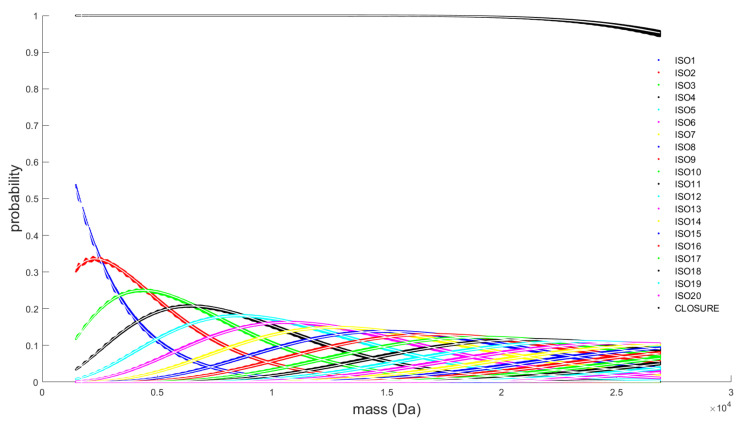
Scatterplot of the first 20 isotopes of all possible DNA molecules within the restricted mass range between 1463.2424 and 26,899.3222 Da. Every isotope variant is denoted by a different colour coding. Given the limited availability of colours in the plot functionality, the coding scheme is repeated, but the context should make it clear which isotopes are being discussed. The plot illustrates how the probability (*y*-axis) for a particular aggregated isotope variant evolves in function of the monoisotopic mass (*x*-axis). The black line on the top of the figure is the coverage that sums the probabilities of the first 20 isotopes per DNA molecule.

**Figure 3 metabolites-11-00400-f003:**
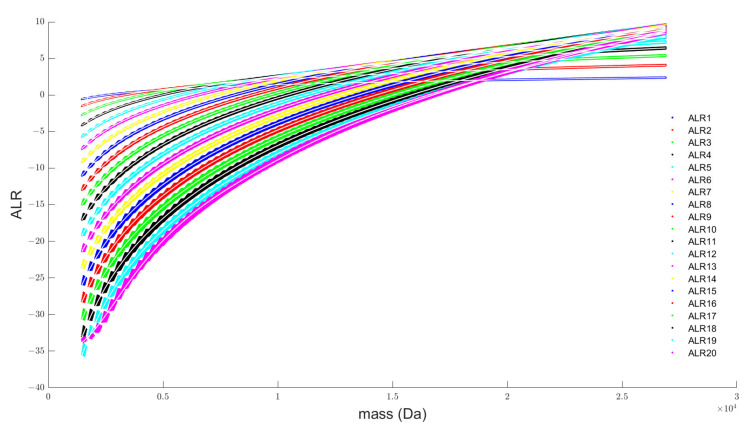
Scatterplot of the ALR transformed isotopes. The monoisotopic variant is taken as the reference isotope for this transformation. ALR20 is the additive log-ratio transformation of the pseudo-isotope derived from the coverage term in Figure 2.

**Figure 4 metabolites-11-00400-f004:**
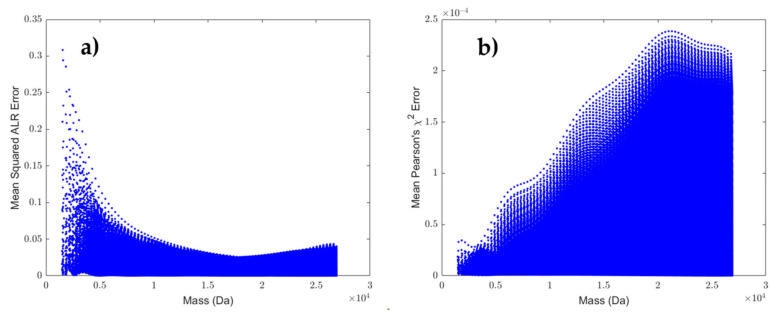
Panel (**a**) provides the mean squared error in the ALR space for the DNA molecules in the restricted mass range. For each of the first 20 isotopes, the squared error is computed between the theoretical ALR transformed isotope and the predicted ALR isotope from the final model. Next, the mean squared error for every DNA molecule is computed by taking the mean of the error over the first 20 isotopes. Panel (**b**) provides a similar graphic, except the error is computed as Pearson’s chi-squared error in the simplex (i.e., probabilities) space.

**Figure 5 metabolites-11-00400-f005:**
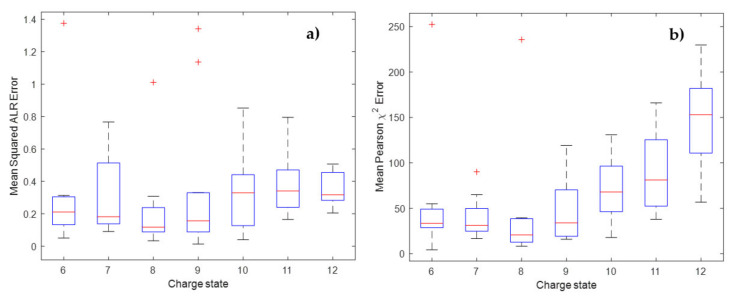
The error distributions for compound DNA_short1 are provided as Tukey’s box and whisker plots across the different charge states. Each box composes 10 repeated measurements of the compound over the liquid chromatography-dimension of the experiment. Panel (**a**) gives the distribution of the mean squared ALR error. Panel (**b**) gives the distribution of the mean Pearson’s chi-squared error.

**Figure 6 metabolites-11-00400-f006:**
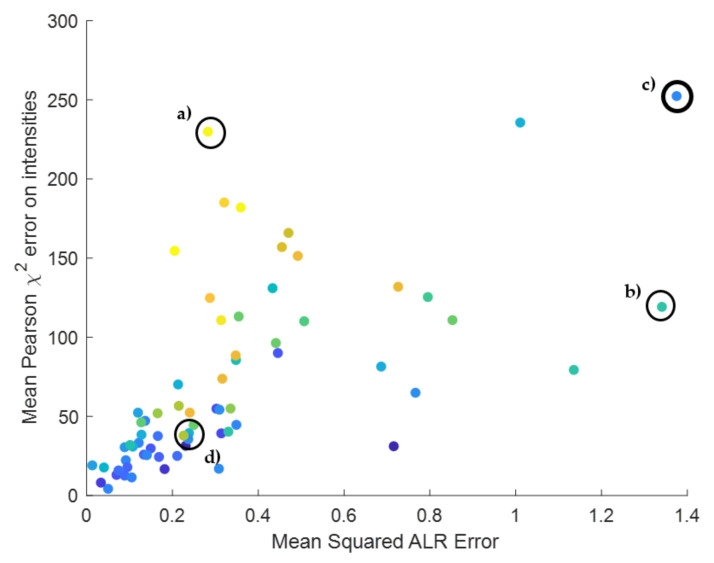
Scatterplot of the MSE in the ALR space (*x*-axis) and the mean Pearson’s chi-squared error on the observed isotopes (*y*-axis). The plot contains 68 points, where each point corresponds to an observed isotope cluster of compound DNA_short1 for which the monoisotopic variant was above the detection limit. The colour coding expresses a third dimension that indicates the log10 of the summed intensities of the isotope cluster. Blue means low intense signal, whilst yellow means high intense signal. The theoretical and observed isotope intensities for the encircled points (**a**,**b**) can be observed in Figure 7a,b respectively. The data for encircled points (**c**,**d**) are provided in Appendix A Appendix A.

**Figure 7 metabolites-11-00400-f007:**
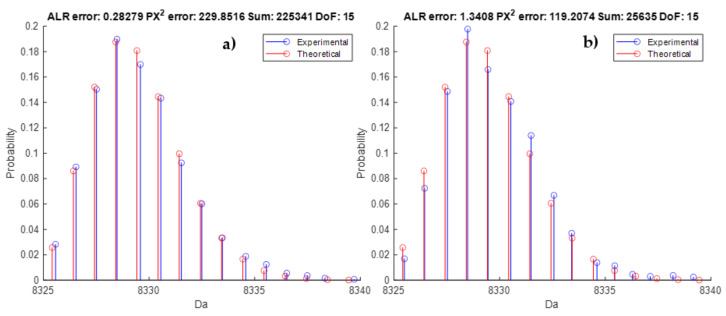
Stem plot illustrating the observed isotope distribution (blue) and theoretical isotope distribution computed by BRAIN using the elemental composition (red). The red lines are the same for both panels. Panel (**a**) is case (**a**) in Figure 6, whilst panel (**b**) is case (**b**). An important remark should be made here with respect to the scaling. In order to keep the *y*-axis comparable across different intensity values, we transform the observed intensities to probabilities. Since the identity of the compound is known, we can also compute/predict the theoretical/predicted probabilities and sum these for the observed aggregated isotope variants. Next, the intensities are scaled to that sum probability. In a sense, this calculation is the reciprocal of the operation specified in Equation (10).

**Figure 8 metabolites-11-00400-f008:**
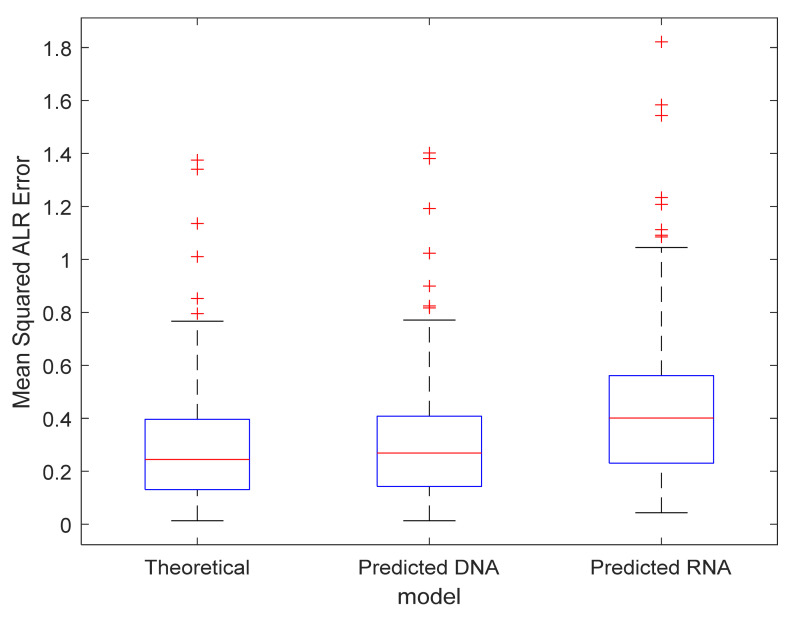
Boxplot of the mean squared ALR error computed with the theoretical model (based on the elemental composition using BRAIN algorithm), predicted with the correct average DNA model and predicted using the misspecified average RNA model. Note that in this visualisation, all the error scores are lumped together disregarding the charge, intensity or the number of peaks in the isotope cluster.

**Figure 9 metabolites-11-00400-f009:**
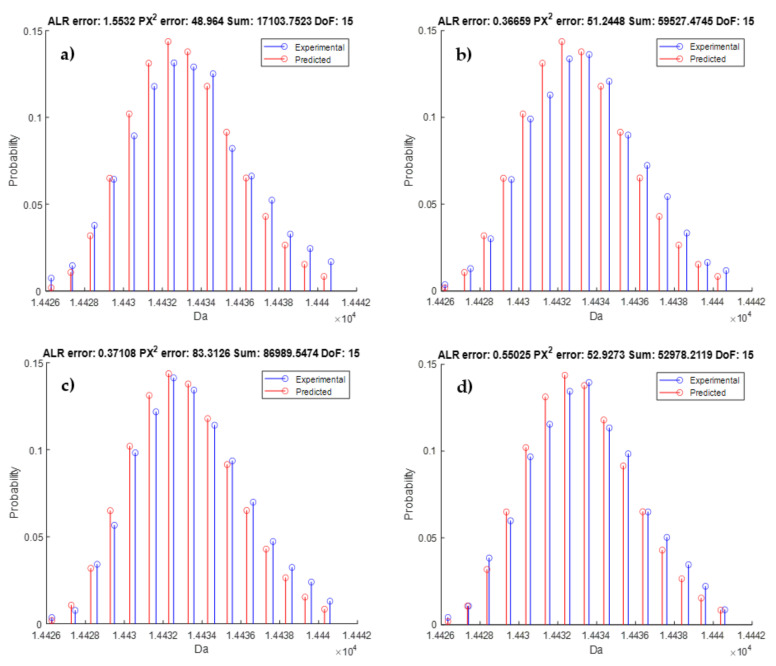
Stem plot illustrating the observed isotope distribution (blue) and predicted isotope distribution from the average DNA model. The red lines are the same for all panels. Panel (**a**–**d**) represent the different charge states of this DNA molecule ranging from z = 15 up to z = 18. The goodness-of-fit statistics in the titles of the subpanels indicate an accurate fit that is in range with previous observations on the smaller DNA molecules.

**Table 1 metabolites-11-00400-t001:** Four strands measured for the proof-of-concept study of isotopic distribution prediction.

Name	DNA_SHORT1	DNA_SHORT2
Type	DNA	DNA
Sequence	GCC ACA TAT GAG AGT GGA TTT GTC ATT	GGT GCC CCA GAA TCT CTC AGC CT
Elemental formula	C266H334N100O162P26	C221H282N82O137P22
Monoisotopic mass	8325.41493	6957.184784
Charge states	6 to 12	5 to 9
Elution ranges	10.95 min–11.05 min (10 scans)	10.32 min–10.45 min (14 scans)
Replicates	7 × 10 = 70	5 × 14 = 70
Name	RNA-like	DNA_long
Type	RNA	DNA
Sequence	As-Afs-Cs-Af-U-Uf-G-A-G-Cf-G-Af-U-Af-U-Cf-C-As-C *[N = 2′OMe, Nf = 2′F, s or PS, phosphorothioate. PO, phosphodiester]*	GAG ATC TCT GCT TCT GAT GGC TCT CTG GTT ACT GCC AGT TGA ATC TG
Elemental formula	C192H239O117N73P18S4F8	C459H582N162O290P46
Monoisotopic mass	6275.90281	14,426.37043
Charge states	5 to 8	15 to 18
Elution ranges	13.75 min–13.85 min (12 scans)	12.1 min–12.3 min (23 scans)
Replicates	4 × 12 = 48	4 × 23 = 92

**Table 2 metabolites-11-00400-t002:** Average mass difference between the centroid masses of the aggregated isotope variants and the monoisotopic variant across the restricted mass range. Isotope 1 is the monoisotopic variant. A mass dependency of the residual can be observed in Appendix A.

DNA				RNA			
Isotope	Mass Difference (Da)	Isotope	Mass Difference (Da)	Isotope	Mass Difference (Da)	Isotope	Mass Difference (Da)
1	0	11	10.02608399	1	0	11	10.02587038
2	1.002707	12	11.02860871	2	1.002698922	12	11.02836784
3	2.005384	13	12.03112309	3	2.005361437	13	12.03085481
4	3.008035	14	13.03362787	4	3.007994300	14	13.03333209
5	4.010663	15	14.03612367	5	4.010602048	15	14.03580039
6	5.013272	16	15.03861108	6	5.013188104	16	15.03826032
7	6.015864	17	16.0410906	7	6.015755128	17	16.04071244
8	7.018439	18	17.0435627	8	7.018305257	18	17.04315722
9	8.021000	19	18.0460278	9	8.020840240	19	18.04559513
10	9.023548	20	19.04848627	10	9.023361538	20	19.04802655

## Data Availability

The data used in this study is available in the Appendix A.

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
