# Peer review of "A Compositional Model to Predict the Aggregated Isotope Distribution for Average DNA and RNA Oligonucleotides"

_metabolites, 2021, doi:10.3390/metabo11060400_

Round 1

Reviewer 1 Report

The manuscript appears well-written, and results are presented in a clear and ordinate way. My main concern is about the simplicity of the model. It would be nice if the authors perform another method (as they did in their previous study; https://pubmed.ncbi.nlm.nih.gov/18325782/) and compare and elaborate the results.

Author Response

The presented research builds further on the mentioned manuscript entitled “A model-based method for the prediction of the isotopic distribution of peptides” by Valkenborg et al.  The advantage of the previous model is that it operates on consecutive isotope ratio’s, making it more robust to noise on a particular peak whilst allowing for an unconstrained polynomial regression approach. Although, the method is still very relevant, it is not capable of predicting the isotope probabilities which are constrained to sum to one and have a value between 0 and 1. A possible solution would be to use models inspired by statistical distributions, such as the Poisson distribution. However, in the manuscript by Valkenborg et al. entitled “Using a Poisson approximation to predict the isotopic distribution of sulphur-containing peptides in a peptide-centric proteomic approach” it is illustrated that such models lack flexibility. In order to obtain predicted probabilities instead of consecutive isotope ratios, we had to revert to a compositional framework that transform the isotopes into log-transformed isotope ratio’s with the monoisotopic variant taken as the reference.  When comparing the operational characteristics of the prediction between consecutive ratio’s and probabilities on our oligo validation dataset, it would require different goodness-of-fit measure and therefore such a comparison would lack an equal basis.  On a more theoretical note, a comparison could be performed between the predicted consecutive isotope ratio’s and isotope ratio’s computed from the predicted probabilities. However, such a comparison would have little practical value as it would focus on differences due to a simple transformation and therefore would make the manuscript overly convoluted and unneeded lengthy.

Reviewer 2 Report

In this manuscript, Annelies A. et al propose a computational model to predict the aggregated isotope distribution of average DNA or RNA molecules. The authors train a univariate weighted polynomial regression model to predict the first 20 isotope peaks for  DNA and RNA. 
The performance of the presented approach was evaluated by comparing it with the experimental data, highlighting the potential of theoretical analysis of the isotopic distribution. Overall, this manuscript is well organized and clearly presented. The only suggestion I have is that the results and discussion should be revised to make the paragraphs more concise and precise focusing on the most relevant topics. For example, on page 9 line 372-374, “Given the limited availability of colours in the plot functionality, the coding scheme is repeated but the context should make it clear which isotopes are being discussed.” could be moved to the figure legend.

Author Response

We want to thank reviewer for these comments. Meanwhile all the co-authors have proofread the manuscript and redundant text has been removed to make the manuscript more in focus and concise. For example, we followed the suggestion of the reviewer and removed the mentioned text to the figure legend. A manuscript version with track changes is uploaded to monitor the text improvements.